# Hierarchical Network-Based Tracklets Data Association for Multiple Extended Target Tracking with Intermittent Measurements

**DOI:** 10.3390/s23146372

**Published:** 2023-07-13

**Authors:** Kaiyi Jiang, Yiguo Li, Tianli Ma, Lin Li

**Affiliations:** School of Electronics and Information Engineering, Xi’an Technological University, Xi’an 710021, China; jiangkaiyi@st.xatu.edu.cn (K.J.); liyiguo@st.xatu.edu.cn (Y.L.); lilin0624@xatu.edu.cn (L.L.)

**Keywords:** multiple extended targets, data association, tracklets, min-cost network flow, intermittent measurements

## Abstract

The key issue of multiple extended target tracking is to differentiate the origins of the measurements. The association of measurements with the possible origins within the target’s extent is difficult, especially for occlusions or detection blind zones, which cause intermittent measurements. To solve this problem, a hierarchical network-based tracklet data association algorithm (ET-HT) is proposed. At the low association level, a min-cost network flow model based on the divided measurement sets is built to extract the possible tracklets. At the high association level, these tracklets are further associated with the final trajectories. The association is formulated as an integral programming problem for finding the maximum a posterior probability in the network flow model based on the tracklets. Moreover, the state of the extended target is calculated using the in-coordinate interval Kalman smoother. Simulation and experimental results show the superiority of the proposed ET-HT algorithm over JPDA- and RFS-based methods when measurements are intermittently unavailable.

## 1. Introduction

Conventional multiple target tracking (MTT) algorithms assume that objects can be represented as points and allow only a single measurement per sensor scan. However, modern high-resolution sensors have revealed the existence of targets that can generate multiple measurements per scan [1]. This challenges the suitability of the conventional point-target assumption. In such scenarios, multiple extended target tracking (METT) provides a more appropriate approach as it specifically addresses the tracking of targets that can produce multiple measurements per scan. The MTT with more measurements per target is called multiple extended target tracking [2,3].

Data association plays a vital role in multi-target tracking by differentiating between false alarms and actual targets. Incorrect data association can significantly impact the performance of tracking multiple targets. For point targets, several methods for data association have been proposed, which can be found in [4]. On the other hand, data association for extended targets involves the challenging task of matching measurements to specific targets, and this complexity increases with the number of targets and measurements. To address this, multiple data association techniques have been developed specifically for multiple extended target tracking scenarios. Vivone introduced a method that incorporates a detector and joint probability data association (JPDA) tracker specifically designed for METT [5]. Additionally, the multi-detection JPDA (MD-JPDA) algorithm was developed to address many-to-one associations in high-resolution radar sensors [6]. However, MD-JPDA suffers from high complexity due to exhaustive combinations of measurements. To mitigate this issue, MD-JIPDA, an extension of MD-JPDA, integrates the existence probability, reducing complexity [7,8]. Another approach by Yang focuses on calculating marginal association probabilities for METT without relying on exhaustive hypotheses and partitions [9]. Furthermore, several algorithms based on the random finite set (RFS) framework have been proposed. These include the probability hypothesis density (PHD) filter [10,11], cardinalized PHD filter [12], generalized labeled multi-bernoulli (GLMB) filter [13], and Poisson multi-bernoulli mixture (PMBM) [14]. These RFS-based methods offer both optimal and suboptimal state estimates for multiple extended targets. However, it is important to note that tuning parameters for RFS-based methods may impact system reliability [15]. In addition, the graphical model formulation is applied to deal with the METT problem. Su presents the belief propagation algorithm to obtain estimators based on a simplified measurement set [16]. In order to remove the deviation in the extended state estimation, a METT algorithm based on Loopy Belief Propagation is presented in [17]. Previously developed METT algorithms rely on continuous measurement availability [18].

Intermittent measurements are a common occurrence in practical applications due to occlusions, detection blind zones (DBZs), and low frame rates caused by radar operation. To handle subsequent miss-detections, Mahdi proposed the Interacting Multiple Model PHD tracker [19], which does not output track labels. Another method, the multiple-model multiple-hypothesis PHD (MM-MH-PHD) filter, adopts a multiple-model approach to estimate motion states in blind zones [20]. In the case of tracking maneuvering targets in blind areas while considering DBZ masking, the MM-GLMB filter is utilized in [21], incorporating a minimum detection speed. These works struggled to handle the METT problem. Hence, Yang performed goal association by combining the PHD filter with the network flow, using a network flow algorithm to solve the minimum cost maximum flow problem, which is able to handle temporary disturbances with good robustness. In Ref. [22], this combination provides a more robust multi-target tracking algorithm that can handle temporary interference and generate longer and more consistent target trajectories.

The algorithms mentioned above require the known intermittent probability when dealing with the tracking process of intermittent measurements. Once the intermittent probability changes, it will affect the tracking results and cause a decrease in tracking accuracy.

In this paper, a hierarchical network framework of the METT with intermittent measurements is proposed. The core of the algorithm lies in the association between the reliable tracklets. A layered network is exploited with respect to the low-level min-cost flow constructed by the clustered measurements set and the high-level min-cost flow constructed by these tracklets. The reliable tracklets can be obtained by using the A* algorithm on the low network. Then, the minimum cost flow algorithm is employed to obtain the trajectories from the directed acyclic graph. The experimental results show that the hierarchical network-based tracklets algorithm (ET-HT) is effective. Our main contributions are as follows: (1) A hierarchical network data association framework for multiple extended target tracking with intermittent measurements is proposed. (2) The min-cost network-based tracklets are built, and the A * search algorithm is used to solve the integer programming problem. (3) The multiple pedestrians tracking scenario is adopted to test the performance of the proposed algorithm.

The structure of this paper is as follows: Section 2 provides the problem statement. Section 3 describes the proposed ET-HT algorithm including the formulation of the point-based and tracklet-based network, and the global optimal solution. In Section 4, the simulation and experimental results are presented to prove the effectiveness of the ET-HT algorithm. Section 5 is the discussion. Section 6 is the conclusion.

## 2. Problem Formulations

The METT problem aims to estimate states and parameters while dealing with measurements from multiple extended targets and the presence of clutter. At each time step *k*, the states of the multiple extended targets are denoted as $S_{k}=\left\{s_{1,k},s_{2,k},..,s_{N_{T},k}\right\}$, $N_{T}$ representing the number of extended targets. Each extended target state is defined as $s_{i,k}={\left[x_{i,k},X_{i,k}\right]}^{T}$. The subset of the state $x_{i,k}$ includes the centroid’s location $m_{i,k}$ and the velocity ${\dot{m}}_{i,k}$ of the centroid of the target, $x_{i,k}={\left[m_{i,k},{\dot{m}}_{i,k}\right]}^{T}$. $X_{i,k}\in R^{d}$ is the extension state that describes the size and shape of the *i*-th target. At the time step *k*, the unordered set of measurements containing the clutter and target $Z_{k}$ is received by the sensor. $Z_{k}=\left\{z_{1,k},z_{2,k},\ldots ,z_{M_{k},k}\right\}$, where $M_{k}=\sum_{i}^{N_{T}}M_{i,k}+M_{C,k}$ represents the number of measurements. $M_{i,k}$ is defined as the total number of measurements that originates from the *i*-th extended target. The value of clutter $M_{C,k}$ is Poisson-distributed with mean $\lambda_{C}$.

In multiple extended target tracking, the measurements may intermittently be unavailable at any time. This is due to occlusions, detection blind zones, or even low frame rates caused by the work mode of the radar. To cater for the missing measurement, a binary variable representing the existence of the extended target is introduced, inspired by [23]. $r_{k}=\left\{r_{1,k},r_{2,k},\ldots ,r_{N_{T},k}\right\}$, with $r_{i,k}\in \left\{0,1\right\}$, where the values of 0 and 1 correspond to the loss of the data and the measurement set of the *i*-th extended target is successfully received, respectively. The time-homogeneous binary Markov process has a transition probability matrix given by
(1)$$\pi^{+}=(\{r_{i,k+1}\;=\;n\left|r_{i,k}\;=\;m\right.\}{)}_{m,n\in S}=\left[\begin{array}{cc} 1-q_{s,i} & q_{s,i}\\ p_{f,i} & 1-p_{f,i} \end{array}\right]$$
where S={0,1} is defined as the state space of the Markov process. The parameters $p_{f,i}$ and $q_{s,i}$ denote the failure rate and recovery rate, $0<p_{f,i},q_{s,i}<1$, such that the Markov process ${\left\{r_{k}\right\}}_{k\geq 0}$ is ergodic. Obviously, a smaller value of $p_{f}$ and a larger value of $q_{s}$ indicate a more reliable measurement received. Thus, the obtained measurement set of the *i*-th extended target at the time step *k* can be described as
(2)$$Z_{i,k}=r_{i,k}{\left\{z_{i,k}\right\}}^{M_{i,k}}$$

The objective of METT is to accurately and robustly track multiple targets, even in challenging scenarios with occlusions, target interactions, intermittent measurements, and other complexities [24].

## 3. Hierarchical Network-Based Data Association for Multiple Extended Target Tracking

In this paper, emphasis is placed on the modeling of the data association for multiple extended targets (METs), utilizing a hierarchical dense neighborhood search approach. To achieve this, the density-based spatial clustering algorithm is employed to divide measurements at a specific time into clusters. Subsequently, a low-level association network is constructed based on the clustering results, enabling the calculation of tracklets. The association of clusters is formulated as a maximum a posteriori (MAP) problem, taking into account target initiation, termination, and false trajectories arising from clutter. At a higher level, the association network estimates trajectories by utilizing the tracklets derived from the low-level network. The paper presents the flowchart of the ET-HT filter in Figure 1, illustrating the stepwise process.

### 3.1. Pre-Processing

Since the set of measurements includes extended targets and clutter, it is necessary to extract the measured data of every extended object in one time step. Considering the property of the measurements of the extended target in that detections are spatially distributed around, the density-based spatial clustering algorithm is used to partition the set of measurements into multiple partitions. For the set of measurements $Z_{k}$ which is received by the sensor at the time *k*, the set of subpartitions is
(3)$$\Pi_{k}=\left(\pi_{1,k},\pi_{2,k},\ldots \pi_{N_{\pi },k}\right)$$
where $\pi_{i,k}$ represents a single subpartition, and $N_{\pi }$ is the number of all subpartitions. Note that the traditional partition algorithms include distance partition, subpartition, and K-means clustering. These methods require the specification of the amount of clusters. The density-based spatial clustering can determine the number of clusters automatically in this paper, based on the intrinsic structure of the set of measurements. The details of the implementation can be found in [25].

### 3.2. Hierarchical Association

*a*.
*Low-level association network*


Due to the intermittent observations in the extended target tracking system, the extended target trajectories may be divided into several unconnected tracklets. Let $T_{i}=\left(\tau_{i}^{1},\tau_{i}^{2},\ldots ,\tau_{i}^{n}\right)$ represent the *i*-th extended target trajectory, where $\tau_{i}^{j}$ is defined as the *j*-th tracklet of the *i*-th target, and the number of trajectories is unknown. $T=(T_{1},T_{2},\ldots ,T_{m})$ is the set of the tracklets. In this section, the set of subpartitions is denoted as $\Pi_{1:k}=\left(\Pi_{1},\Pi_{2},\ldots ,\Pi_{k}\right)$. The low-level association defines $\Pi_{1:k}$ as the input, and uses the network flow method to generate the tracklets. The key here is to calculate a MAP estimate for T with a set of a cluster of measurements $\Pi_{1:k}$.
(4)$$T^{*}=\arg\max\prod_{T}P\left(T|\Pi_{1:k}\right)$$

In this chapter, the G(V,E) is defined as a directed acyclic graph, where V represents the set of nodes and E represents the set of edges, with *s* denoting the nodes of the graph and *n* being the set of edges of the graph. This gives the set of graph nodes V and E, with $v_{i}$ defined as a subpartition and each edge $e_{j}$ in G(V,E) representing the motion between the subpartitions. In solving the data association process, the concept of network flow is used to represent $f_{ij}$ as a directed flow variable from node $v_{i}$ to node $v_{j}$, where $f_{si}$ and $f_{jn}$ denote the starting flow variable and the ending flow variable, respectively. Each flow in the graph is subject to the following constraints. Firstly, the sum of the flows arriving at node $v_{i}$ is equal to the sum of the flows leaving this node at the same moment. For any tracklet $\tau_{i}^{j}$:(5)$$f_{si}+\sum_{j:ji\in E}f_{ji}=\sum_{j:ij\in E}f_{ij}+f_{jn}$$

Secondly, the cost flow network must ensure that nothing other than a single extended object can be represented at one time. The upper bound of the sum of outgoing flows from node $v_{i}^{}$ is set to 1. For any node:(6)$$\forall v_{i},v_{j}\sum f_{ij}<1$$

Considering that targets can appear or disappear anywhere in the cost-flow network, a source node and a sink node are introduced in [26], which are connected to the respective nodes. The source and sink nodes are also subject to a constraint that enforces all the flows starting in *s* to end in *n*.
(7)$$\sum_{i}f_{si}=\sum_{i}f_{in}$$

Through the network optimization process, Equation (Equation 4) is transformed into an integer programming (IP) problem, and the logarithm of the objective function is given by
(8)$$\begin{array}{cc} T^{*} & =\arg\max\prod_{T}P\left(T|\Pi_{1:k}\right)\\ \; & =\arg\max\sum_{T}c_{si}f_{si}+c_{ij}f_{ij}+c_{jn}f_{jn}\\ \; & =\arg\max\sum_{i}c_{si}f_{si}+\sum_{ij}c_{ij}f_{ij}+\sum_{j}c_{jn}f_{jn} \end{array}$$
where $c_{si}$ is the flow cost from the source node to the subpartition $\pi_{i}$, $c_{ij}$ is the flow cost from the subpartition $\pi_{i}$ to the subpartition $\pi_{j}$, and $c_{jn}$ is the flow cost from the subpartition $\pi_{j}$ to the sink node. Hence, the IP problem can be described as
(9)$$\begin{array}{c} \min\sum_{i}c_{si}f_{si}+\sum_{ij}c_{ij}f_{ij}+\sum_{j}c_{jn}f_{jn}\\ s.t.\forall \pi_{i},\pi_{j}\sum f_{ij}<1\\ \forall \pi_{i},\pi_{j}f_{i,j}\geq 0 \end{array}$$

The cost can be defined as follows:(10)$$\left\{\begin{array}{c} c_{si}=-\log P_{s}\left(\pi_{i}\right)\\ c_{ij}=-\log P_{l}\left(\pi_{j}|\pi_{i}\right)\\ c_{jn}=-\log P_{n}\left(\pi_{j}\right) \end{array}\right.$$

In this paper, the data association problem is transformed into a MAP estimation for T. The A * search algorithm is used to solve the problem, and the optimum trajectory $T^{*}$ is obtained.

*b*.
*High-level association network*


In this section, the tracklet-based network flow framework is represented as $G^{*}\left(V^{*},E^{*}\right)$ to obtain the trajectories of the multiple extended targets. It can be also formulated as a MAP problem.
(11)$$\begin{array}{cc} \Psi^{*} & =\underset{\Psi^{{}^{\prime }}}{\arg\max}P\left(\Psi^{{}^{\prime }}|T^{*}\right)\\ \; & =\underset{\Psi^{{}^{\prime }}}{\arg\max}P\left(T^{*}|\Psi^{{}^{\prime }}\right)P\left(\Psi^{{}^{\prime }}\right)\\ \; & =\underset{\Psi^{{}^{\prime }}}{\arg\max}\prod_{T_{i}^{*}\in T^{*}}P(T_{i}^{*}|\Psi^{{}^{\prime }})\prod_{\Psi_{j}\in \Psi^{{}^{\prime }}}P(\Psi_{j}) \end{array}$$
where $\Psi_{i}$ denotes the merged trajectory, $\Psi_{i}=\left\{T_{i,0}^{*},T_{i,1}^{*},\ldots ,T_{i,l}^{*}\right\}$. *l* denotes the number of tracklets in $\Psi_{i}$, and $\Psi^{{}^{\prime }}=\left\{\Psi_{i}\right\}$ is the merged trajectories set.

Note that the set of linear constraints is similar to those of Equations (Equation 5)–(Equation 7). The only difference is in the tracklet-based network flow model, where each node represents a tracklet extracted in the low-level association stage, which is a set of continuous measurements in a batch of time frames.

Assuming that the likelihoods of the input tracklets are conditionally independent given the merged trajectories set, and each merged trajectory is independent, the cost flow network of the tracklets is given by
(12)$$P\left(\Psi_{j}\right)=P_{\Psi }\left(T_{i,0}^{*}\right)\ldots P_{l}\left(T_{i,l-1}^{*}|T_{i,l}^{*}\right)P_{t}\left(T_{i,l}^{*}\right)$$
where $P(\Psi_{j})$ represents a Markov Chain. $P_{s}\left(T_{i,0}^{*}\right)$, $P_{l}\left(T_{i,l-1}^{*}|T_{i,l}^{*}\right)$ and $P_{t}\left(T_{i,l}^{*}\right)$ represent the initial probability, transition probability, and terminal probability, respectively.

The link cost between two tracklets is defined as
(13)$$P_{l}\left(T_{i}^{*}|T_{j}^{*}\right)=p_{m}\left(T_{i}^{*}|T_{j}^{*}\right)p_{t}\left(T_{i}^{*}|T_{j}^{*}\right)$$
where $p_{m}\left(T_{i}^{*}|T_{j}^{*}\right)$ and $p_{t}\left(T_{i}^{*}|T_{j}^{*}\right)$ represent the motion cost and time cost, respectively. The motion-associated probability is defined as
(14)$$p_{m}\left(T_{i}^{*}|T_{j}^{*}\right)=-\ln\left(N\left(\Delta p_{1},\Sigma \right)N\left(\Delta p_{2},\Sigma \right)\right)$$
(15)$$\Delta p_{1}\;=\;p_{i}^{tail}+v_{i}^{tail}\Delta t-p_{j}^{head}$$
(16)$$\Delta p_{2}\;=\;p_{j}^{head}-v_{i}^{head}\Delta t-p_{i}^{tail}$$
where Δt denotes the frame gap between the last detection set of the tracklet $T_{j}^{*}$ and the first detection set of the tracklet $T_{i}^{*}$. $p_{i}^{head}$ and $p_{i}^{tail}$ are the center position of the first and last detection set of the tracklet $T_{i}^{*}$, respectively. $v_{i}^{head}$ and $v_{i}^{tail}$ indicate the estimated speed of the tracklet $T_{i}^{*}$ at the head and tail, respectively. The motion affinity between two tracklets is shown in Figure 2.

In this case, it is assumed that the error of the predicted location and the central location of the detection set $\Delta p_{1}$ and $\Delta p_{2}$ follows a Gaussian distribution. The smaller the error between the predicted location and the actual position of the target to be connected, the greater the motion similarity between the corresponding track slices will be. The temporal associated cost is defined as
(17)$$p_{t}\left(T_{i}^{*}|T_{j}^{*}\right)\;=\;\left\{\begin{array}{c} 1,\\ 0, \end{array}\begin{array}{c} \Delta t\in [1,\varsigma ]\\ otherwise \end{array}\right.$$
where ς is the upper bound of the frame gap. The initialization probabilities and termination probabilities for each tracklet are set to be
(18)$$P_{\Psi }\left(T_{i}^{*}\right)=P_{t}\left(T_{j}^{*}\right)=1$$

Similar to the low-level association process, the network flow model based on tracklets is established to solve the motion cost and time cost between the tracklets, and the optimal trajectories are obtained using the A* search algorithm.

### 3.3. Trajectory Smoothness

In this section, the in-coordinate interval Kalman smoothing algorithm is used to calculate the extended target’s state[27]. The algorithm consists of forward filtering and backward smoothing. Assume that the interval between two frames is defined as k(i)=k(0)+i×T, $\left(i=0,1,2,\cdots ,M_{l}\right)$, where *T* is the sampling time. At the time $k(i)(i=0,1,2,\cdots ,M_{l}-1)$, the Kalman filter only predicts the state of the extended target until a new measurement arrives at time $k(M_{l})$ [28]. The time update step is as follows:(19)$$s_{k(i)|k(i-1)}=F_{k(i-1)}s_{k(i-1)}^{}$$
(20)$$P_{k(i)|k(i-1)}=F_{k(i-1)}P_{k(i-1)}F_{k(i-1)}^{T}+w_{k(i)}$$
where $s_{k(i)}$ denotes the extended target state at the time k(i), and $F_{k(i-1)}$ is the transition matrix. Once the measurements arrive at time $k(M_{l})$, the forward filtering step is
(21)$$s_{k\left(M_{l}\right)|k\left(M_{l}-1\right)}=F_{k(M_{l}-1)}s_{k(M_{l}-1)}$$
(22)$$P_{k\left(M_{l}\right)|k\left(M_{l}-1\right)}=F_{k(M_{l}-1)}P_{k(M_{l}-1)}F_{k(M_{l}-1)}^{T}+w_{k(M_{l})}$$
(23)$$K_{k(M_{l})}=P_{k\left(M_{l}\right)|k\left(M_{l}-1\right)}H_{k(M_{l})}^{T}{\left[H_{k(M_{l})}P_{k\left(M_{l}\right)|k\left(M_{l}-1\right)}H_{k(M_{l})}^{T}+e_{k(M_{l})}\right]}^{-1}$$
(24)$$s_{k(M_{l})}^{}=s_{k\left(M_{l}\right)|k\left(M_{l}-1\right)}+K_{k(M_{l})}\left[{\bar{z}}_{k(M_{l})}-H_{k(M_{l})}s_{k\left(M_{l}\right)|k\left(M_{l}-1\right)}\right]$$
(25)$$P_{k(M_{l})}=\left[I-K_{k(M_{l})}H_{k(M_{l})}\right]P_{k\left(M_{l}\right)|k\left(M_{l}-1\right)}$$
where ${\bar{z}}_{k(M_{l})}$ represents the central location of the measurements of the extracted extended target at time $k(M_{l})$. Here, the state and covariance matrix at each time are calculated via the backward recursive method starting from the last time $k_{max}$, given by
(26)$$s_{k|k}=F_{k}s_{k|k+1}$$
(27)$$P_{k|k}=F_{k}P_{k|k+1}F_{k}^{T}$$
(28)$$J_{k}=P_{k|k+1}F_{k+1}^{T}P_{k|k+1}^{-1}$$
(29)$$s_{k|k_{max}}=s_{k|k}+J_{k}\left(s_{k+1|k_{max}}-s_{k|k+1}\right)$$
(30)$$P_{k|k_{max}}=P_{k|k}+J_{k}\left(P_{k+1|k_{max}}-P_{k|k+1}\right)J_{k}^{T}$$
where $s_{k|k_{max}}$ and $P_{k|k_{max}}$ represent the smoothness state estimation and covariance matrix at time *k*, respectively. $J_{k}$ is the smoothness gain matrix.

## 4. Numerical Simulation and Experiments

In this section, the feasibility of the proposed algorithm is verified by numerical simulation and experimental verification.

### 4.1. Case 1: Numerical Simulation

In the simulation, the validity of the ET-HT filter is tested. Consider a 2D surveillance region which is set as [3000, 10,000] m × [0,6000] m, with a clutter intensity of $c_{k}=5\times 10^{-17}$. The time step is set to 30, and the sampling period is defined as T=1 s. There are two extended targets, and their initial positions are set to [6000,4500] m and [3500,3000] m, respectively. Their start velocities are set to [0,−150] m/s and [200,−20] m/s, respectively. The start time and terminal time in this system are [1,5] s and [5,30] s, respectively. The kinematic state is given by
(31)$$s_{i,k}=F_{i,k}s_{i,k-1}+w_{i,k}$$
where $s_{i,k}=\left[x_{i,k},y_{i,k},{\dot{x}}_{i,k},{\dot{y}}_{i,k}\right]$ is the state variable. $x_{i,k}$ and $y_{i,k}$ represent the location of the targets. ${\dot{x}}_{i,k}$ and ${\dot{y}}_{i,k}$ represent the velocity of the targets. $F_{i,k}$ is the kinematic state transition matrix of the *i*-th target. $w_{i,k}$ represents the Gaussian process noise of the *i*-th target with zero mean and covariance $Q_{i,k}$, $Q_{i,k}=diag\left[100,100,0,0\right]$.

The measurement model is defined as
(32)$$z_{i,k}=H_{k}s_{i,k}+e_{i,k}$$
where $H_{k}$ is the observation matrix, $z_{i,k}$ is defined as the measurements generated by the *i*-th target at time *k*, and $e_{i,k}$ denotes the Gaussian measurement noise of the *i*-th target with zero mean and covariance $R_{i,k}$, $R_{i,k}=diag\left[100,100\right]$.

In the simulation, the clutter Poisson rate $\lambda_{c}$ is set to 600, and then the clutter density $\lambda_{c}c_{k}$ is $3\times 10^{-15}$. The failure and recovery rate are set to $p_{f}=0.2$ and $q_{s}=0.8$, respectively. The simulated target scenario with intermittent measurements is shown in Figure 3.

In the experiment, the absolute mean number of targets estimation error (NTE) [29] and the optimal subpattern assignment (OSPA) distance [30] are taken into account as metrics to evaluate the performance of the proposed ET-HT filter against the ET-PHD, ET-JPDA, ET-PMBM, and ET-NFPHD filters.

The absolute mean number of targets estimation error is defined as
(33)$$NTE\left(X_{k},Y_{k}\right)=E\{|X_{k}\left|-\right|Y_{k}\left|\right\}$$
where $X_{k}$ and $Y_{k}$ are two finite subsets, and $|X_{k}|$ and $|Y_{k}|$ represent the potential of the two subsets, respectively. The OSPA distance between $X_{k}$ and $Y_{k}$ is defined as the distance between the position and the potential of the two sets.
(34)$$OSPA\left(p,c,X_{k},Y_{k}\right)={\left(\frac{1}{n}\left(\min_{\pi \in \prod {}_{n}}\sum {}_{i=1}^{n}d^{\left(c\right)}{\left(x_{i},y_{i}\right)}^{p}+c^{p}\left(m-n\right)\right)\right)}^{1/p},n\leq m$$
where *c* is the penalty cost for the cardinality mismatch, and *p* is the order of the OSPA metric, 1<p<∞, c>0. In the simulation, they are set to 10 and 2, respectively.

The OSPA distances and NTEs of the five filters are depicted in Figure 4. Comparatively, the ET-HT filter exhibits a lower OSPA distance and NTEs when compared to the other filters. In contrast, the performance of the ET-JPDA filter is influenced by clutter and necessitates prior knowledge regarding the number of targets. As the estimation error increases, the OSPA distance and NTE of the ET-JPDA filter also escalate. On the other hand, the ET-PHD filter leverages the first-order approximate moment of the multi-target density to convey valuable information about target potential. When the motion trajectories intersect, the targets are partitioned into one cell, which leads to an increase in the OSPA distance. Meanwhile, the ET-PHD and ET-PMBM filters require precise models of both targets and clutter. However, the lack of prior information and the interference of clutter result in a large discrepancy between the estimated and actual number of targets, which incurs increases in the NTE. The ET-NFPHD filter utilizes the PHD filter to process sensor measurements and extract the PHD component at different time steps. These PHD components are integrated into the network flow graph as nodes. By solving the minimum cost maximum flow problem among these nodes, the optimal data association path can be determined. As a result, the ET-NFPHD filter demonstrates a reduced OSPA distance and NTE compared to the ET-PHD filter. However, due to the direct modeling of the PHD filter’s update data, the algorithm suffers from accumulated update errors, resulting in inferior tracking performance compared to the ET-HT filter proposed in this study.

In order to illustrate the influence of clutter on the experiment, we designed simulation experiments with different clutter numbers.

Figure 5 presents the average OSPA distances and NTEs of the five filters at varying clutter rates. The ET-HT filter consistently demonstrates smaller average OSPA distance and NTEs compared to the ET-JPDA, ET-PMBM, ET-PHD and ET-NFPHD filters. To delve further into the influence of intermittent probability, the investigation extends to a different failure rate $p_{f}$ and recovery rate $q_{s}$. The OSPA distances and NTEs of the five filters are illustrated in Figure 6a,b, respectively, considering diverse recovery rates in 100 Monte Carlo runs. Likewise, Figure 7a,b depicts the OSPA distances and NTEs of the filters for varying failure rates with 100 Monte Carlo runs.

The results from Figure 6 and Figure 7 clearly demonstrate that the ET-HT filter exhibits superior tracking performance compared to the ET-PHD, ET-PMBM, ET-JPDA, and ET-NFPHD filters irrespective of the transformation of the measurement recovery and loss rates. This is because the ET-HT filter associates the tracklets into long tracks through both low-level and high-level associations, thereby reducing the impact of intermittent measurements.

On the other hand, the experimental results of the ET-JPDA filter show that the OSPA distance of the ET-JPDA filter tends to decrease as the measurement recovery rate $q_{s}$ increases, as shown in Figure 6a. However, if the measurement loss rate $p_{f}$ increases, as shown in Figure 7a, it may result in a mismatch between the tracking gate and the measurement, thereby increasing the OSPA distance. Similarly, the results obtained using the ET-PHD filter shown in Figure 6b and Figure 7b demonstrate that with an increasing recovery rate, some pieces of clutter are incorrectly identified as targets, causing errors in target estimation and an increase in the NTE. Conversely, a higher loss rate $p_{f}$ leads to greater data loss, resulting in a decrease in the effective data volume at each time step. In the ET-PHD filters, significant data loss implies that fewer measurement data are associated with targets, leading to a decrease in the estimated number of targets and NTE. The presence of clutter impacts the partitioning of measurements, causing the OSPA distance and NTE of the ET-PMBM filter to exceed those of the ET-HT filter. The depicted results in Figure 6 and Figure 7 demonstrate the consistent superiority of the ET-NFPHD filter over the ET-PHD filter, irrespective of variations in intermittent probability. These findings serve as evidence supporting the notion that the integration of the flow network method effectively enhances the accuracy of the PHD filter.

### 4.2. Time Complexity

The time complexity of the ET-HT filter consists of two main components, namely the adaptive spectral clustering algorithm and the worst-case scenario of the search subgraph. The time complexity of the former is $O\left(n\times \left(N_{tc}-n_{tc}\right)\right)+O\left(n_{tc}^{3}\right)+O\left(K_{tc}N_{tc}I_{tc}\right)\cdot O\left(n_{tc}\times \left(N_{tc}-n_{tc}\right)\right)$, where $K_{tc}$ is the number of nodes in the undirected graph, $n_{tc}$ is the sample points, $N_{tc}$ is the number of trajectories, and $I_{tc}$ is the iteration number of the K-means algorithm. The worst time complexity of the searching sub-graph is $O(n_{\max}^{2})$ [31].

The time complexity of the ET-JPDA filter is $O(N_{tc}\times n_{tc}^{2})$, where $N_{tc}$ is the initial number of targets, and $n_{tc}$ is the sampling time. The time complexity of the ET-PHD filter is $O\left(N_{tc}\times n_{tc}^{2}\right)+O\left(n_{tc}\times M_{tc}\times N_{tc}^{2}\times d_{tc}^{2}\right)$, where the parameter $M_{tc}$ is generally greater than 1000, and $d_{tc}$ is the dimension of the state vector. The time complexity of the ET-PMBM filter is $O(N_{tc}^{3}\times H_{tc}^{3})$, where $H_{tc}$ represents the number of hypothetical combinations. The time complexity of the ET-NFPHD filter is $O\left(N_{tc}\times n_{tc}^{2}\right)+O\left(K_{tc}N_{tc}I_{tc}\right)$.

The proposed algorithm and existing algorithms are coded with MatlabR2019b, and the experiments are run on a computer with an Intel Core i74770k CPU at 3.5 GHz. The running times of the five filters for 100 Monte Carlo runs are shown in Table 1. 

Table 1 demonstrates that the ET-HT filter exhibits a reduced running time in comparison to both the ET-PHD filter and the ET-PMBM filter. This indicates that the ET-HT filter possesses a lower time complexity, leading to significant enhancements in computational efficiency. Although the running time of the ET-HT filter exceeds that of the ET-JPDA filter, its accuracy in estimating the state and number of targets is significantly superior. Hence, it is crucial to prioritize accuracy in tracking rather than solely emphasizing the attainment of low time complexity.

### 4.3. Case 2: Real Data Verification

To evaluate the applicability of the proposed filter, case 2 incorporates real scenarios. These selected scenes encompass more randomness and a higher number of moving targets, reflecting real situations. The positional data of each pedestrian were obtained with a pedestrian detector with discriminatively trained part-based models [32]. Firstly, a classifier that uses a decision rule learned from the data was constructed to determine whether the image window contains pedestrians. Next, the image was divided into multiple sub windows and each sub window was described using features that were invariant to lighting effects. Then, each sub window was provided to the classifier, which determined whether the sub-window contained pedestrians, and their position coordinates were extracted.

The input for the ET-HT filter consisted of a collection of measurement points gathered from all frames during the motion sequence. In situations where pedestrians were obscured by a streetlight, the measurement points could not be extracted, thereby simulating intermittent observation scenarios.

The ET-HT filter framework was applied to a scenario from the PETS 2009S0CC dataset. In this dataset, we focused on a specific scene featuring six pedestrians walking. Their paths intersected at specific frames (295, 305, 345, 365, and 367). Additionally, in frames 319, 339, and 351, the pedestrians were obstructed by a street light, as illustrated in Figure 8a. Figure 8b shows the extracted measurements. Figure 9 shows the tracking results of the ET-HT filter. Following a similar approach as in case 1, the PETS2009S0CC dataset was utilized to evaluate the performance of the ET-HT filter, ET-JPDA filter, ET-PHD filter, ET-PMBM, and ET-NFPHD filters of a single example run. The OSPA distances and NTEs were computed for each filter with the time step t=10 s. The obtained results are illustrated in Figure 10.

As illustrated in Figure 9, the ET-HT filter achieves a better performance when dealing with real data with intermittent measurements and is able to accurately estimate the trajectories of the targets. The results in Figure 10a,b show that the tracking error of the ET-JPDA filter increases with the increase in clutter number, while the ET-PHD, ET-PMBM, and ET-NFPHD filters have large errors in dealing with real scenes. On the contrary, the ET-HT filter has better robustness and accuracy for tracking targets in real scenes.

The validation of the ET-HT filter on real data demonstrates its advantages in addressing target tracking problems with intermittent measurements. These findings have significant implications for practical applications that demand precise target tracking, such as intelligent traffic systems, video surveillance, and object recognition.

## 5. Discussion

The primary focus of this paper is to tackle the challenge of tracking multiple extended targets when dealing with intermittent measurements. The proposed filter in this study, ET-HT, is designed to extract all potential short tracks and employs a network flow model to compute the trajectories of these targets. Comparative experiments have been conducted to validate the algorithm’s performance, demonstrating its superior robustness and accuracy in handling intermittent measurements.However, it is important to note that this paper exclusively addresses the tracking method for extended targets and does not explore the issue of tracking group targets with intermittent measurements. Additionally, the implications of target separation and merging phenomena on tracking performance in the presence of intermittent measurements have not been thoroughly investigated in this paper. Further exploration of this aspect is warranted and represents a crucial area for future research.

## 6. Conclusions

In this paper, we present a hierarchical network-based tracklet data association framework for tracking multiple extended targets with intermittent measurements. At low association levels, all possible tracklets are extracted and utilized to construct a high-level network flow model. The trajectories of the multiple extended targets can then be obtained using the A* search algorithm. The effectiveness of the proposed ET-HT filter is demonstrated through simulations and experimental results, particularly in challenging scenarios involving clutter, newborn targets, and occlusion. Moving forward, our future research aims to expand the algorithm’s capabilities by considering observation delay, integrating it into multi-object trackers, and exploring the potential of incorporating data from multiple sensors to enhance estimation accuracy and overall tracking performance.

## Figures and Tables

**Figure 1 sensors-23-06372-f001:**
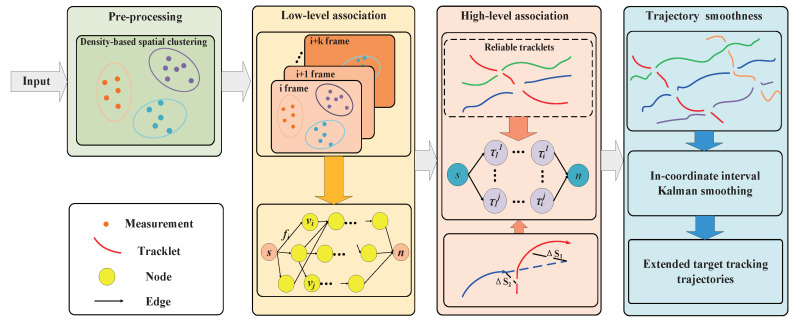
The flowchart of the proposed algorithm.

**Figure 2 sensors-23-06372-f002:**
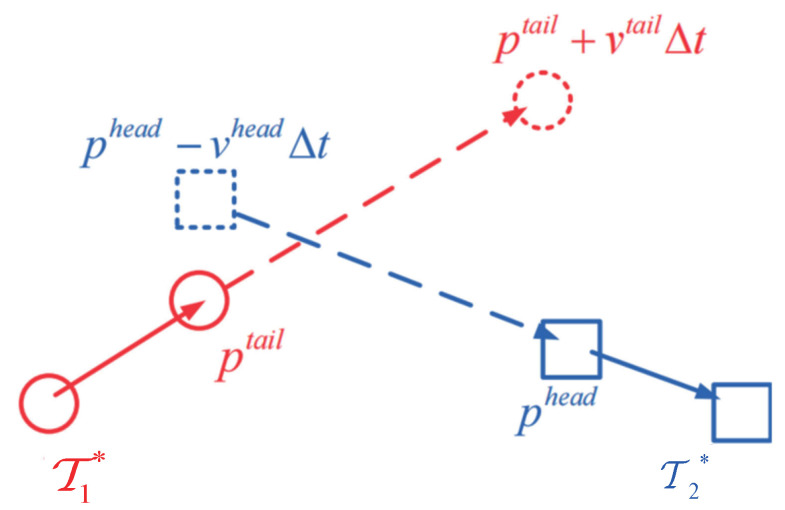
The motion affinity between two tracklets.

**Figure 3 sensors-23-06372-f003:**
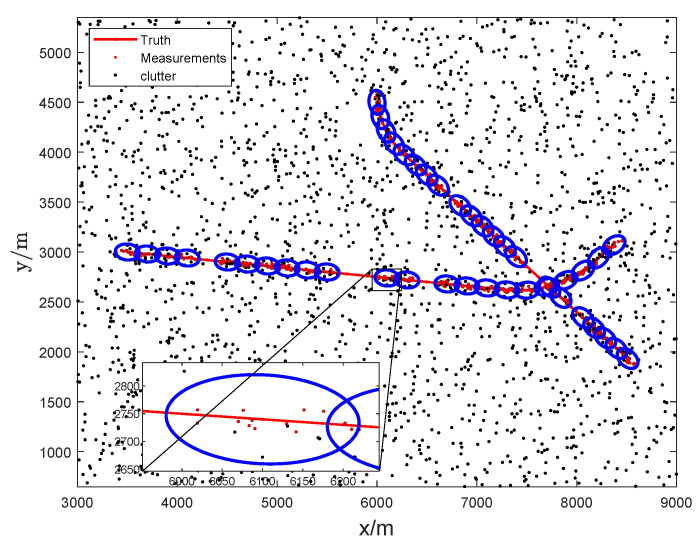
Target trajectories and measurements, the blue ellipses represent the shape of the extended targets.

**Figure 4 sensors-23-06372-f004:**
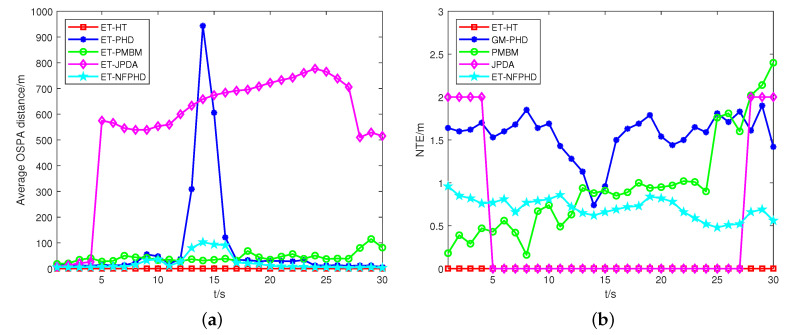
The average OSPA distances and NTEs of a single example run, with $q_{s}=0.8$, $p_{f}=0.2$, $\lambda_{C}=300$. (**a**) OSPA, (**b**) NTE.

**Figure 5 sensors-23-06372-f005:**
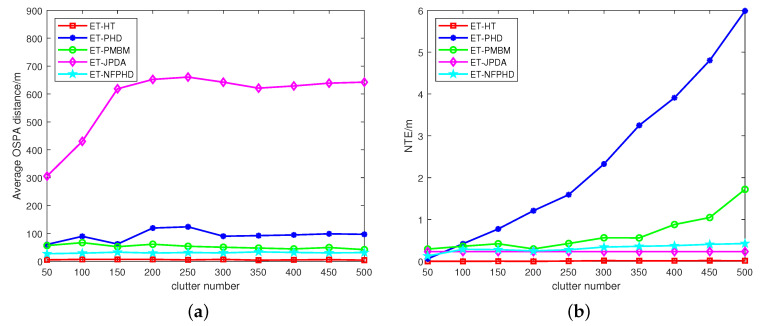
The average OSPA distances and NTEs for 100 Monte Carlo runs, with $q_{s}=0.8$, $p_{f}=0.2$. (**a**) OSPA, (**b**) NTE.

**Figure 6 sensors-23-06372-f006:**
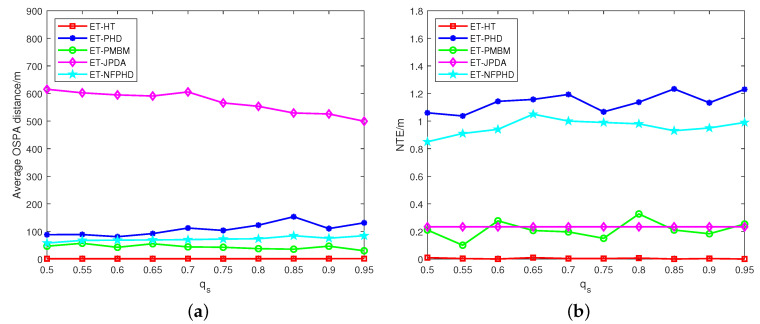
The average OSPA distances and NTEs of different recovery rates for 100 Monte Carlo runs, with $p_{f}=0.2$, $\lambda_{C}=600$. (**a**) OSPA, (**b**) NTE.

**Figure 7 sensors-23-06372-f007:**
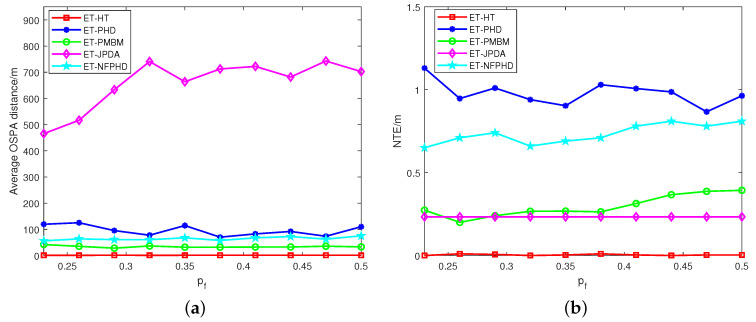
The average OSPA distances and NTEs of different failure rates for 100 Monte Carlo runs, with $q_{s}=0.8$, $\lambda_{C}=600$. (**a**) OSPA, (**b**) NTE.

**Figure 8 sensors-23-06372-f008:**
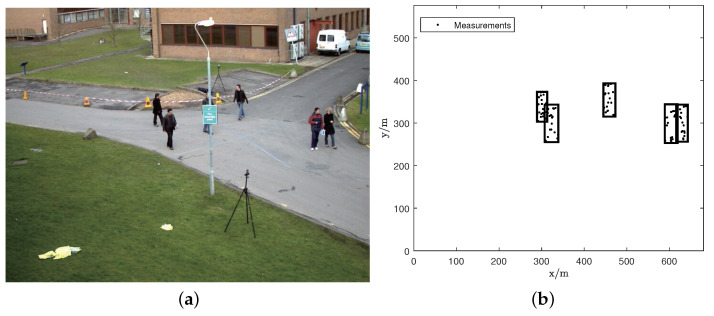
An example of the PETS2009S0CC dataset. The extracted measurement points are fed into the ET-HT filter. Subsequently, following processing with the density-based spatial clustering algorithm, these measurement points can be considered as trajectories depicting pedestrian motion. (**a**) An example of the PETS 2009S0CC dataset where a pedestrian is obstructed by a street light, and (**b**) the extracted measurements.

**Figure 9 sensors-23-06372-f009:**
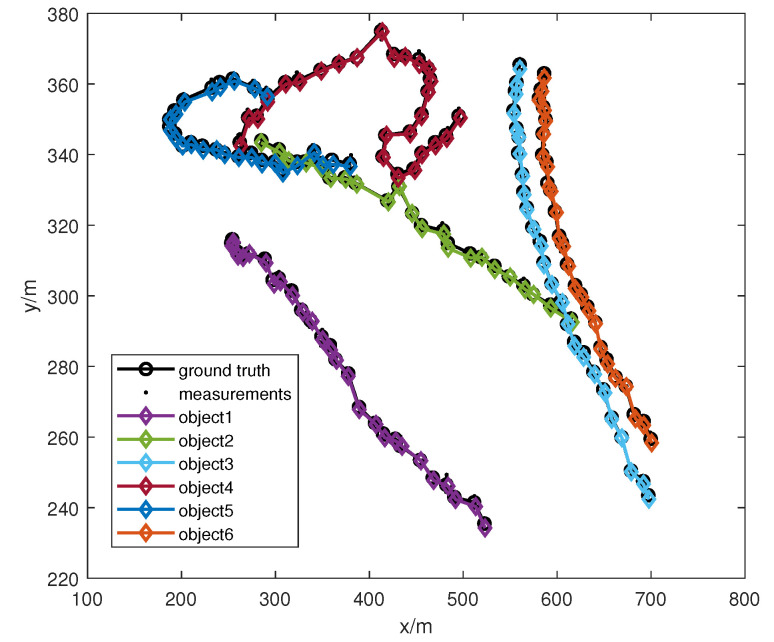
The tracking results for the ET-HT filter.

**Figure 10 sensors-23-06372-f010:**
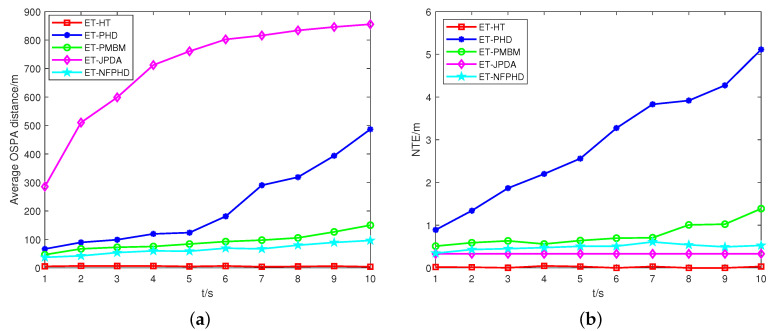
The average OSPA distances and NTEs of a single example run, with t=10. (**a**) OSPA, (**b**) NTE.

**Table 1 sensors-23-06372-t001:** Average running times of five filters.

Filter	ET-HT	ET-PHD	ET-PMBM	ET-JPDA	ET-NFPHD
Time (t/s)	10.1369	2232.9	96.3917	1.1254	411.9493

## Data Availability

The data based on School of Electronics and Information Engineering is not universal, so it is not disclosed. Interested readers can contact us directly by email.

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
