# Peer review of "Hierarchical Network-Based Tracklets Data Association for Multiple Extended Target Tracking with Intermittent Measurements"

_sensors, 2023, doi:10.3390/s23146372_

Round 1
Reviewer 1 Report
1. The author needs to provide a more detailed analysis of the experimental results in Figures 6-7.
2. The introduction of the hardware platform needs to be placed before table 1.
3. What is the biggest difference between Case 2 and Case 1?
4. The abstract needs to be modified to allow others to quickly know what you are doing.
Moderate editing of English language required
Author Response
Response to Reviewer 1 Comments
Point 1: The author needs to provide a more detailed analysis of the experimental results in Figures 6.
Response 1: Thanks for your comments. We have provided a more detailed analysis of the experimental results in Figure 6. We have added the OSPA distances and NTEs of the proposed algorithm, ET-JPDA, ET-PHD, ET-NFPHD and ET-PMBM. The explanations have been added to illustrate the estimation accuracy of the proposed algorithm. (Please see Line 271-285, Page 9-10)
Point 2: The introduction of the hardware platform needs to be placed before table 1.
Response 2: Thanks for your comments on our paper. The introduction of the hardware platform was modified according to the comment. (Please see Line 298-300, Page 12)
Point 3: What is the biggest difference between Case 2 and Case 1?
Response 3: Thanks for your comment. In case 1, we simulated the scene of target motion by constructing the motion model and measurement model. In case 2, we extract the coordinates of targets in a real motion scene using the pedestrian detector with discriminatively trained part-Based models, which serves as input to the ET-HT algorithm to verify the effectiveness of the proposed algorithm.
Point 4: The abstract needs to be modified to allow others to quickly know what you are doing
Response 4: Thanks for your comment. Based on your suggestions, we have rewritten the abstract as follows:
“The key issue of the multiple extended targets tracking is to differentiate the origins of the measurements. The association of measurements to the possible origins within the target’s extent is difficult, especially for the occlusions or the detection blind zone which cause the intermittent measurements. To solve the problem, a hierarchical network-based tracklets data association algorithm (ET-HT) is proposed. At the low association level, a min-cost network flow model based on the divided measurement sets is built to extract the possible tracklets. At the high association level, these tracklets are further associated to form final trajectories. The association is formulated as an Integral Programming problem for finding the maximum a posterior probability in the network flow model based on the tracklets. Moreover, the state of the extended target is calculated by the in-coordinate interval Kalman smoother. Simulation and experimental results show the proposed ET-HT algorithm superiority over JPDA and RFS-based methods when measurements are intermittently unavailable.”

Reviewer 2 Report
This article aims to address multiple challenges in extended object tracking, particularly in scenarios where occlusions or detection blind spots lead to intermittent measurements. The authors propose a trajectory data association algorithm based on a hierarchical network to extract potential trajectories and utilize the A* search algorithm to compute the optimal trajectories for multiple extended objects. Additionally, Kalman filtering is introduced to perform forward filtering and backward smoothing on the trajectories, thereby obtaining smooth state estimations. The algorithm demonstrates good performance in simulation experiments, with minimal sensitivity to clutter parameters and intermittent probabilities.
The overall article is well-structured and has practical significance. However, there are still some areas for improvement:
1. Figures 5(a) / 6(a) / 7(a) lack the unit "m" on the y-axis.
2. The description of case2 is rather brief and not detailed enough to be convincing. Suggest a more detailed description of case2.
3. In Figure 8(a), pedestrians are not obstructed by streetlights, and it is recommended to replace the figure. Additionally, an explanation should be provided regarding how the measurements in Figure 8(b) are extracted.
4. Figure 9 lacks labels for the coordinate axes, and it is advisable to use "x/m" and "y/m" consistently.
5. On page 12, in the Conclusion section, the third and fourth lines mentioning "A*" should appear in the same line.
Author Response
Response to Reviewer 2 Comments
Point 1: Figures 5(a)/ 6(a)/7(a)lack the unit "m" on the y-axis
Response 1: We are so sorry for our carelessness. The unit "m" on the y-axis have been added into the revised manuscript. (Please see Figure5, Page 10; Figure 6, Page 11; Figure7, Page11).
Point 2: The description of case2 is rather brief and not detailed enough to be convincing. Suggest a more detailed description of case2.
Response 2: We are grateful for the suggestion. To be clearer and in accordance with the reviewer concerns, we have added the description of case 2 as follows:
“The input for the ET-HT filter consisted of a collection of measurement points gathered from all frames during the motion sequence. In situations where pedestrians were obscured by a streetlight, the measurement points could not be extracted, thereby simulating intermittent observation scenarios.
Following a similar approach as in case 1, the PETS2009S0CC dataset was utilized to evaluate the performance of the ET-HT filter, ET-JPDA filter, ET-PHD filter, ET-PMBM and ET-NFPHD filters of a single example run. The OSPA distances and NTEs were computed for each filter with time steps t=10s. The obtained results are illustrated in Figure 10”. (Please see Line 316-318, Page 12; Line 323-325, Page 12; Figures10(a) and10(b), Page13).
Point 3: In Figure 8(a), pedestrians are not obstructed by streetlights, and it is recommended to replace the figure. Additionally, an explanation should be provided regarding how the measurements in Figure 8(b) are extracted.
Response 3: Thank you for your suggestion. We have replaced the Figure 8(a), in which the pedestrians are obstructed by streetlights. We also have explanted the measurement extracted process as follow:
“The positional data of each pedestrian was obtained by the pedestrian detector with discriminatively trained part-Based models. Firstly, construct a classifier that is a decision rule learned from the data to determine whether the image window contains pedestrians. Next, divide the image into multiple sub windows and describe each sub window using features that are invariant to lighting effects. Then, each sub window is provided to the classifier, which will determine whether the sub window contains pedestrians and extract their position coordinates.”. (Please see Line 310-315, Page 12; Figure8(a), Page 13).
Point 4: Figure 9 lacks labels for the coordinate axes, and it is advisable to use "x/m" and "y/m" consistently.
Response 4: We are so sorry for our carelessness. The labels "x/m" and "y/m" have been added into the revised manuscript. (Please see Figure 9, Page 13).
Point 5: On page 12, in the Conclusion section, the third and fourth lines mentioning "A*should appear in the same line.
Response 5: Thanks for your comments on our paper. The conclusion has been modified as follow:
“In this paper, we present a hierarchical network-based tracklets data association framework for tracking multiple extended targets with intermittent measurements. At the low association levels, all possible tracklets are extracted and utilized to construct a high-level network flow model. The trajectories of the multi-extended targets can then be obtained using the A* search algorithm. The effectiveness of the proposed ET-HT filter is demonstrated through simulations and experimental results, particularly in challenging scenarios involving clutter, new-born targets, and occlusion. Moving forward, our future research aims to expand the algorithm's capabilities by considering observation delay, integrating it into multi-object trackers, and exploring the potential of incorporating data from multiple sensors to enhance estimation accuracy and overall tracking performance.” (Please see Line349-357, Page 14).

Reviewer 3 Report
1. Object tracking is one of the leading scientific problems of recent years. Both in civilian and military applications. This paper presents aspects of tracking for multiple extended targets with intermittent measurements.
2. General remarks
a. Too many abbreviations make it difficult to follow the content of the article. Each abbreviation should be expanded the first time it appears. Not all readers need to know all abbreviations.
b. Please use the language of a scientific research report without personal references: “we”, “our”, which are many in whole article.
c. The subject of object tracking is commonly associated with radar tracking, also using artificial neural networks. It is worth to compare algorithms used in radar tracking to the solution proposed in the paper. Publications worth analyzing include: doi: 10.2478/v10177-011-0009-8
d. However the article is well written should be carefully edited. Some remarks included below.
3. Specific remarks
a. There is lack of “Discussion” chapter.
b. The axes in Figure 9 should be labeled.
c. The final conclusions are too general and only generally summarize the research presented in the article. I suggest expanding the conclusions with more detailed findings.
Author Response
Response to Reviewer 3 Comments
Point 1: Too many abbreviations make it difficult to follow the content of the article. Each abbreviation should be expanded the first time it appears. Not all readers need to know all abbreviations.
Response 1: Thank you. We are so sorry for our carelessness. The missing definitions have been added into the revised manuscript. Such as
The word“MTT”was corrected as“Multiple Targets Tracking(MTT)”. (Please see Line 16, Page 1)
The word“METT”was corrected as“Multi-extended Targets Tracking(METT)”. (Please see Line 20, Page 1)
The word“IP problem”was corrected as“Integer programming (IP) problem”. (Please see Line 148, Page 5)
The word“OSPA distance”was corrected as“optimal subpattern assignment (OSPA) distance”. (Please see Line 231, Page 9)
Point 2: Please use the language of a scientific research report without personal references:“we".“our", which are many in whole article.
Response 2: Thanks for your comment. We revised the manuscript by using the language of a scientific research report. Such as
The sentence “In this section, we use the in-coordinate interval Kalman smoothing algorithm to calculate the extended target’s state.” was corrected as “In this section, the in-coordinate interval Kalman smoothing algorithm is used to calculate the extended target's state.”(Please see Line 188-189, Page 7)
The sentence “In our experiment, we take into account the…” was corrected as “In the experiment, the absolute mean number of targets estimation error (NTE) and the optimal subpattern assignment (OSPA) distance are taken into account”(Please see Line 230-231, Page 9)
The sentence “We input the extracted measurement points into the ET-HT filter.” was corrected as “The extracted measurement points are fed into the ET-HT filter.”(Please see the caption of figure 8, Page 13)
The sentence “we demonstrate the advantages of the ET-HT filter in solving
target tracking problems with intermittent measurements” was corrected as “The validation of the ET-HT filter on real data demonstrates its advantages in addressing target tracking problems with intermittent measurements.”(Please see Line 333-334, Page 14)
Point 3: The subject of object tracking is commonly associated with radar tracking, also using artificial neural networks. It is worth to compare algorithms used in radar tracking to the solution proposed in the paper. Publications worth analyzing include: doi: 10.2478/10177-011-0009-8
Response 3: The paper given by the reviewer proposes a method for radar data fusion in multi-sensor radar systems, mainly introducing the application of target tracking in radar monitoring systems, as well as algorithms and methods for multi-sensor data fusion. The article mainly discusses two methods of multi-sensor data fusion: decentralized fusion and centralized fusion, and analyzes in detail the advantages and disadvantages of decentralized fusion.
Our paper presents a novel approach called the Hierarchical Network-based Tracklets Data Association algorithm, designed specifically for multiple extended target tracking with intermittent measurements.
The two papers focus on different fields of research, but the paper provided by the reviewer has great research value in the field of multi target tracking in multi-sensor data fusion. We cited it in the introduction section.
Point 4: There is lack of “Discussion” chapter.
.
Response 4: We are grateful for the suggestion. To be clearer and in accordance with the reviewer concerns, we have added a "Discussion" chapter. The Discussion is as follows:
“The primary focus of this paper is to tackle the challenge of tracking multiple extended targets when dealing with intermittent measurements. The proposed filter in this study, ET-HT, is designed to extract all potential short tracks and employ a network flow model to compute the trajectories of these targets. Comparative experiments have been conducted to validate the algorithm's performance, demonstrating its superior robustness and accuracy in handling intermittent measurements. However, it is important to note that this paper exclusively addresses the tracking method for extended targets and does not explore the issue of tracking group targets with intermittent measurements. Additionally, the implications of target separation and merging phenomena on tracking performance in the presence of intermittent measurements have not been thoroughly investigated in this paper. Further exploration of this aspect is warranted and represents a crucial area for future research.”. (Please see Line 338-347, Page 14)
Point 5: The axes in Figure 9 should be labeled.
Response 5: We are so sorry for our carelessness. The axes have been added in Figure 9. (Please see Figure 9, Page 13)
Point 6: The final conclusions are too general and only generally summarize the research presented in the article. I suggest expanding the conclusions with more detailed findings.
Response 6: We are grateful for the suggestion. To be clearer and in accordance with the reviewer concerns, we have rewritten the conclusion as follows:
“In this paper, we present a hierarchical network-based tracklets data association framework for tracking multiple extended targets with intermittent measurements. At the low association levels, all possible tracklets are extracted and utilized to construct a high-level network flow model. The trajectories of the multi-extended targets can then be obtained using the A* search algorithm. The effectiveness of the proposed ET-HT filter is demonstrated through simulations and experimental results, particularly in challenging scenarios involving clutter, new-born targets, and occlusion. Moving forward, our future research aims to expand the algorithm's capabilities by considering observation delay, integrating it into multi-object trackers, and exploring the potential of incorporating data from multiple sensors to enhance estimation accuracy and overall tracking performance.”(Please see Line 349-357, Page 14)

Reviewer 4 Report
Attach the review file.

Author Response
Response to Reviewer 4 Comments
Point 1: Rewrite the abstract in a concise manner.
Response 1: We are grateful for the suggestion. To be clearer and in accordance with the reviewer concerns, we have rewritten the abstract as follows:
“The key issue of the multiple extended targets tracking is to differentiate the origins of the measurements. The association of measurements to the possible origins within the target’s extent is difficult, especially for the occlusions or the detection blind zone which cause the intermittent measurements. To solve the problem, a hierarchical network-based tracklets data association algorithm (ET-HT) is proposed. At the low association level, a min-cost network flow model based on the divided measurement sets is builded to extract the possible tracklets. At the high association level, these tracklets are further associated to form final trajectories. The association is formulated as a Integral Programming problem for finding the maximum a posterior probability in the network flow model based on the tracklets. Moreover, the state of the extended target is calculated by the the in-coordinate interval Kalman smoother. Simulation and experimental results show the proposed ET-HT algorithm superiority over JPDA and RFS-based methods when measurements are intermittently unavailable.”
Point 2: The manuscript needs comprehensive English language proofreading, there are many linguistic mistakes.
Response 2: Thanks for your comment. We have carefully revised the manuscript according to the reviewers' comments, and also have re-scrutinized to improve the English.
The sentence “Then, the minimum cost flow algorithm is employed to get the long trajectory from the directed acyclic graph.” was corrected as “Then, the minimum cost flow algorithm is employed to get the trajectories from the directed acyclic graph.”(Please see Line 66-67, Page 2)
The sentence “Each flow in the graph is subject to the following constraints, firstly, the sum of the flows arriving at node.. Scendly…” was corrected as “Each flow in the graph is subject to the following constraints. Firstly, the sum of the flows arriving at node.. Scendly…”(Please see Line 138-139, Page 5)
The sentence “the in-coordinate interval Kalman smoothing algorithm is chosen to calculate the extended target's state.” was corrected as “the in-coordinate interval Kalman smoothing algorithm is used to calculate the extended target's state.”(Please see Line 188-189, Page 7)
The sentence “to obtain the full trajectory of the extended target.” was corrected as “to obtain the trajectories of the multi-extended targets.”(Please see Line 157-158, Page 6)
The sentence “The objective of METT is to calculate the maximum probability data association with intermittent measurements, and recursively estimate the multi-target state given a set of observation..” was corrected as “The objective of METT is to accurately and robustly track multiple targets, even in challenging scenarios with occlusions, target interactions, intermittent measurements, and other complexities.”(Please see Line 100-101, Page 3)
Point 3: What is the main contribution of the paper?
Response 3: Thanks for your comments on our paper. Our main contributions are: Our main contributions are: (1) a hierarchical network data association framework for multiple extended targets tracking with intermittent measurements is proposed. (2) The min-cost network-based tracklets is built, and the A * search algorithm is used to solve the Integer Programming problem. (3) The multiple pedestrians tracking scenario is adopted to test the performance of the proposed algorithm. (Please see Line 68-72, Page 2 )
Point 4: The introduction does not provide sufficient background information for readers not in the immediate field to understand the problem/hypotheses.
.
Response 4: Thank you for your comments. The research objective of the article is to solve the problem of tracking multiple extended targets with discontinuous measurements. Traditional data association algorithms have some limitations when dealing with multiple extended target tracking problems with discontinuous measurements. Therefore, this study aims to propose a new data association algorithm that can better handle these challenges. The background information has been added in the introduction. (Please see Line 62-72, Page 2)
Point 5: The literature survey is not balanced, important studies are not cited in this work.
Response 5: Thank you for pointing out our mistake. We have added some references with updated bibliographical review. Please see references 3,18,22,24 and 32 in the References Section.
% Reference 3
Stateczny, Andrzej; Kazimierski, Witold (2011). Multisensor Tracking of Marine Targets - Decentralized Fusion of Kalman and Neural Filters. International Journal of Electronics and Telecommunications, 57(1), –.
% Reference 18
Liu W, Zhu S, Wen C, et al. Structure modeling and estimation of multiple resolvable group targets via graph theory and multi-Bernoulli filter[J]. Automatica, 2018, 89: 274-289.
% Reference 22
Yang S, Teich F, Baum M. Network flow labeling for extended target tracking PHD filters[J]. IEEE Transactions on Industrial Informatics, 2019, 15(7): 4164-4171.
% Reference 24
Liu Z, Ji L, Yang F, et al. Cubature information Gaussian mixture probability hypothesis density approach for multi extended target tracking[J]. IEEE Access, 2019, 7: 103678-103692.
%Reference 32
Felzenszwalb P F, Girshick R B, McAllester D, et al. Object detection with discriminatively trained part-based models[J]. IEEE transactions on pattern analysis and machine intelligence, 2009, 32(9): 1627-1645.
Point 6: I see the motivations for this study need to be made clearer.
Response 6: Thanks for your comment. The motivation for the study is “Traditional algorithms requires a known intermittent probability when dealing with the tracking process of intermittent measurements.”We added the motivation in the introduction. (Please see Line 60, Page 2 )
Point 7: The study objectives must be clearly defined.
Response 7: We are grateful for the suggestion. The objective of this paper is “to solve the problem of multi extended target tracking with time-varying intermittent probabilities.” (Please see Line 60-61, Page 2)
We modified the abstract as follows:
“The key issue of the multiple extended targets tracking is to differentiate the origins of the measurements. The association of measurements to the possible origins within the target’s extent is difficult, especially for the occlusions or the detection blind zone which cause the intermittent measurements. To solve the problem, a hierarchical network-based tracklets data association algorithm (ET-HT) is proposed. At the low association level, a min-cost network flow model based on the divided measurement sets is builded to extract the possible tracklets. At the high association level, these tracklets are further associated to form final trajectories. The association is formulated as a Integral Programming problem for finding the maximum a posterior probability in the network flow model based on the tracklets. Moreover, the state of the extended target is calculated by the the in-coordinate interval Kalman smoother. Simulation and experimental results show the proposed ET-HT algorithm superiority over JPDA and RFS-based methods when measurements are intermittently unavailable.”
Point 8: The results are not clearly explained and must be presented in an appropriate format.
Response 8: We have presented the experimental results through evaluation metrics. In case 1, we used an ET-HT filter to track moving targets and compared the corresponding evaluation metrics with the other four algorithms. The running times of the five algorithms were also presented in a table format. In case 2, we used an ET-HT filter to track a set of scenes in the PETS2009 dataset. Figures 9 and 10 show the tracking results.
Point 9: The findings are not properly described in the context of the published literature.
Response 9: Thanks for your suggestion. We have added references and analyzed them in the introduction the references are as follows:
Reference 3
Stateczny A; Kazimierski W. Multisensor Tracking of Marine Targets - Decentralized Fusion of Kalman and Neural Filters. International Journal of Electronics and Telecommunications, 2011, 57(1), 65–70.
Reference 18
Liu W, Zhu S, Wen C, et al. Structure modeling and estimation of multiple resolvable group targets via graph theory and multi-Bernoulli filter[J]. Automatica, 2018, 89: 274-289.
Reference 22
Yang S, Teich F, Baum M. Network flow labeling for extended target tracking PHD filters[J]. IEEE Transactions on Industrial Informatics, 2019, 15(7): 4164-4171.
Point 10: No significant limitations are discussed. It may be worthwhile to mention the tradeoffs involved in your work.
Response 10: Thanks for your comment. And we have added the Discussion, which mention the tradeoffs involved in this work. The discussion is as follows: “The primary focus of this paper is to tackle the challenge of tracking multiple extended targets when dealing with intermittent measurements. The proposed filter in this study, ET-HT, is designed to extract all potential short tracks and employ a network flow model to compute the trajectories of these targets. Comparative experiments have been conducted to validate the algorithm's performance, demonstrating its superior robustness and accuracy in handling intermittent measurements. However, it is important to note that this paper exclusively addresses the tracking method for extended targets and does not explore the issue of tracking group targets with intermittent measurements. Additionally, the implications of target separation and merging phenomena on tracking performance in the presence of intermittent measurements have not been thoroughly investigated in this paper. Further exploration of this aspect is warranted and represents a crucial area for future research.” (Please see Line 338-347, Page 14 )
Point 11: No comparisons with other works are made. Make comparative simulations with other works (2019-2023).
Response 11: Thanks for your comment. After analyzing and consulting the literature in recent years, we quote the algorithm ' Network Flow Labeling for Extended Target Tracking PHD Filters (ET-NFPHD) ' proposed by Yang in 2019, and test it in case1 and case2. The tracking results are compared with the ET-HT and other filters, and the experimental results are analyzed and summarized. (Please see Figure 5, Page 10;figure6, Page 11;Figure7 Page13)
Analysis and summary include:
The ET-NFPHD filter utilizes the PHD filter to process sensor measurements and extract the PHD component at different time steps. These PHD components are integrated into the network flow graph as nodes. By solving the minimum cost maximum flow problem among these nodes, the optimal data association path can be determined. As a result, the ET-NFPHD filter demonstrates reduced OSPA distance and NTE compared to the ET-PHD filter. However, due to the direct modeling of the PHD filter's update data, the algorithm suffers from accumulated update errors, resulting in inferior tracking performance compared to the ET-HT filter proposed in this study. (Please see Line 250-256, Page 9)
The depicted results in Figure 6 and 7 demonstrate the consistent superiority of the ET-NFPHD filter over the ET-PHD filter, irrespective of variations in intermittent probability. These findings serve as evidence supporting the notion that the integration of the Flow network method effectively enhances the accuracy of the PHD filter. (Please see Line 282-285, Page 10)
Point 12: Conclusion part is very short, should be support main findings of the research. The Conclusion does not provide any concluding remarks or discuss the broader implications of the research. It would be beneficial to summarize the significance of the findings and how they contribute to the fields of Hierarchical network-based tracklets data association.
Response 12: We are grateful for the suggestion. To be more clear and in accordance with the reviewer concerns, we have rewritten the conclusion as follows :
“In this paper, we present a hierarchical network-based tracklets data association framework for tracking multiple extended targets with intermittent measurements. At the low association levels, all possible tracklets are extracted and utilized to construct a high-level network flow model. The trajectories of the multi-extended targets can then be obtained using the A* search algorithm. The effectiveness of the proposed ET-HT filter is demonstrated through simulations and experimental results, particularly in challenging scenarios involving clutter, new-born targets, and occlusion. Moving forward, our future research aims to expand the algorithm's capabilities by considering observation delay, integrating it into multi-object trackers, and exploring the potential of incorporating data from multiple sensors to enhance estimation accuracy and overall tracking performance.” (Please see Line 349-357, Page 14)

Round 2
Reviewer 2 Report
The authors have addressed my concerns.
Reviewer 4 Report
The authors addressed all my comments to improve the paper's quality and readability. The academic Editor may consider to the publication if other reviewers also suggested for acceptance.
English is fine.